# Matrices Activated with Messenger RNA

**DOI:** 10.3390/jfb14010048

**Published:** 2023-01-15

**Authors:** Raquel Martinez-Campelo, Marcos Garcia-Fuentes

**Affiliations:** BiDD Group, CiMUS Research Center, Department of Pharmacology, Pharmacy and Pharmaceutical Technology and Health Research Institute of Santiago de Compostela (IDIS), University of Santiago de Compostela, 15782 Santiago de Compostela, Spain

**Keywords:** mRNA-activated matrix, tissue engineering, mRNA delivery, bone regeneration, immunomodulation

## Abstract

Over two decades of preclinical and clinical experience have confirmed that gene therapy-activated matrices are potent tools for sustained gene modulation at the implantation area. Matrices activated with messenger RNA (mRNA) are the latest development in the area, and they promise an ideal combination of efficiency and safety. Indeed, implanted mRNA-activated matrices allow a sustained delivery of mRNA and the continuous production of therapeutic proteins in situ. In addition, they are particularly interesting to generate proteins acting on intracellular targets, as the translated protein can directly exert its therapeutic function. Still, mRNA-activated matrices are incipient technologies with a limited number of published records, and much is still to be understood before their successful implementation. Indeed, the design parameters of mRNA-activated matrices are crucial for their performance, as they affect mRNA stability, device immunogenicity, translation efficiency, and the duration of the therapy. Critical design factors include matrix composition and its mesh size, mRNA chemical modification and sequence, and the characteristics of the nanocarriers used for mRNA delivery. This review aims to provide some background relevant to these technologies and to summarize both the design space for mRNA-activated matrices and the current knowledge regarding their pharmaceutical performance. Furthermore, we will discuss potential applications of mRNA-activated matrices, mainly focusing on tissue engineering and immunomodulation.

## 1. Introduction

Biocompatible matrices have been used for years in tissue engineering to harness the natural regenerative capacity of the organism [1]. These systems create 3D environments and provide the biophysical and biochemical cues necessary to regulate cell behaviour [2,3,4]. Apart from acting as a support, they can also deliver therapeutic molecules [2]. Some of the first molecules to be used were conventional drugs [5,6] and recombinant proteins [1,7]. For example, in the 1990s, different growth factors were embedded in matrices for osteochondral regeneration [1,7]. However, recombinant protein therapy is expensive [8] and high doses are needed to offset the short half-life of these proteins in vivo. Thus, side effects such as ectopic callus formation can appear [9].

To achieve a prolonged but transient protein expression, plasmid DNA (pDNA)-activated matrices were developed by the mid-1990 [10]. In these technologies, pDNA is condensed into vehicles, which are subsequently introduced into the matrices used for implantation. Then, the pDNA is released in a spatiotemporally controlled manner, inducing cell transfection, and finally, the production of target proteins [11]. On the heels of this technology, matrices were also activated with viral vectors. Virus activation soon gained traction and these matrices were then explored in several applications, even reaching the clinical setting [12,13]. Despite being promising techniques, the risk of genomic integration and carcinogenesis, as well as the low transfection efficiency observed in those prototypes, led to the exploration of RNA-activated matrices [14,15]. 

MicroRNAs (miRNAs) [16,17] and small interfering RNAs (siRNAs) [18,19] were the first types of therapeutic RNA to be integrated into implantable matrices. Those share some similarities, but miRNAs can upregulate or downregulate different gene sets, while siRNAs act by silencing specific genes. Both have been incorporated into various scaffolds such as hydrogels, collagen scaffolds, and hydroxyapatite scaffolds for bone regeneration [14]. Long noncoding RNAs have also been used for matrix activation, although less research has been done with this prototype variety [14]. 

In 2015, Elangovan et al. were the first to demonstrate the osteogenic potential of mRNA-activated matrices [20]. These systems, also called transcript-activated matrices (TAMs), showed higher transfection efficiencies, lower toxicity, and more extended protein expression than pDNA-activated matrices [21,22]. Another advantage of using mRNAs is that their production is simple and can be performed in vitro. In addition, the use of non-viral vectors for mRNA delivery reduces the risk of unwanted immune responses [15]. The construction of TAMs and the transfection process that triggers after their implantation is schematized in Figure 1.

These properties have led TAMs to be investigated for promoting tissue formation, as well as for cell differentiation and reprogramming [22,23,24]. Using the administration of mRNAs coding for transcriptional factors, adult cells can be reprogrammed to a stem cell-like state. Then, these cells can be differentiated into a target cell line to repair a specific tissue [25]. 

Moreover, TAMs are being explored for cancer treatment. Although the release of chemotherapeutic drugs from matrices has been studied [26], recent efforts are focused on the use of TAMs as mRNA vaccines with promising results [27,28,29]. 

However, these delivery platforms still have some limitations. One of the most important is that transfection reagents are required to enhance therapy effectiveness, and carriers with suitable efficacy/toxicity ratios are not available for all applications. In addition, it is not clear whether TAMs are better than pDNA or virus-activated matrices when very prolonged effects are required. Furthermore, mRNA complexes could be unstable after vacuum- or freeze-drying processes, two techniques that might be important to guarantee the long-term stability of TAMs, and consequently, favour their large-scale production.

Regardless of the field, TAM design is an important step. The immunogenicity and short- and long-term stability of the mRNA, the cytotoxicity and efficacy of the gene vectors, as well as the controlled release of the mRNA from the matrix, are just some of the aspects that can be improved with an adequate design. Therefore, we will discuss different formulations that have been tested and summarize which parameters can be modified to enhance their performance (Figure 2). Specifically, we will focus on differences in matrix composition, transfection vectors, and mRNA modifications.

## 2. TAM Design Considerations

### 2.1. Differences between 2D and 3D Transfection

It is known that the dimensionality of the culture can affect the internalization of the mRNA vehicles [30], the transfection, and the protein expression [31,32]. In particular, 3D matrices offer some advantages in comparison with transfection in 2D cultures. 

For instance, Balmayor et al. [32] observed that 3D transfection produces greater amounts of protein than transfection of cell monolayers [32]. Furthermore, protein expression is more constant and prolonged over time [31]. Hence, it is not surprising to observe greater osteogenic potential and increased mineralization when bone morphogenic protein (BMP) mRNAs are incorporated into 3D matrices in comparison to 2D transfection [32]. The higher transfection efficiency observed in these systems could be due to the increased cell viability. Whereas matrices allow a sustained release of the nanoplexes, in 2D transfection, the nanoplexes are released at once into the cells, which leads to increased cytotoxicity [32].

There are also differences in the mechanisms that regulate 2D and 3D transfection. Dhaliwal et al. investigated the endocytosis pathways used to internalize nanoplexes and the role of cytoskeletal dynamics and RhoGTPase-mediated signalling in both types of transfections [30]. They observed that the internalization of nanoplexes in 2D cultures only occurred by caveolae-mediated and clathrin-mediated endocytosis, while in 3D cultures, macropinocytosis was also involved. As for cytoskeletal dynamics, they determined that actin and microtubule polymerization, as well as actin–myosin interactions, were necessary to mediate efficient gene transfer in 3D systems but not in 2D ones. Furthermore, they also found differences in the effect of Rho GTPases. This family of G signalling proteins regulates cytoskeletal dynamics, cell migration, and vesicle trafficking. Unlike in 2D cultures, activation of ROCK, a downstream effector of Rho, is a necessary step for efficient gene transfer in 3D matrices.

Based on these results, cell adhesion to the matrix may influence gene transfer in 3D systems since it mediates the communication between the cell and its environment and, consequently, RhoGTPase activation and cytoskeletal dynamics [30]. Therefore, matrix features that affect cell adhesion (e.g., stiffness or focal adhesion components) could be relevant to achieve efficient 3D gene transfer. Finally, considering that 2D and 3D transfection is regulated by different mechanisms, caution is required when translating transfection protocols from 2D to 3D environments.

### 2.2. Composition and Mechanical Properties of Matrices

Matrix design is one of the most important factors in achieving effectiveness and low cytotoxicity with TAM therapy. The performance parameters are not only influenced by matrix composition but also by its structure, including the mesh size [31,32,33,34]. 

In terms of composition, different polymers, both natural and synthetic, have been explored in the creation of TAMs [35]. In addition, several groups have tested material combinations to modify matrix characteristics such as electrostatic charge or stiffness [36,37]. For example, changes in matrix charge influence their affinity for mRNA and, consequently, can affect its release [36]. Furthermore, the composition and porosity of the matrix have an impact on cell adhesion and proliferation, thus influencing its degradability, mRNA release, and transfection effectiveness [22,32,34,38,39]. Another factor that affects mRNA delivery is the distribution of cargo within the matrix [32]. When the load is located superficially, it is quickly released by diffusion. Conversely, when the load is also trapped inside the matrix, the release is biphasic: a first burst release occurs by diffusion and a prolonged release occurs by matrix degradation [32]. This mechanism explains that, for example, mRNA contained in the pores of calcium phosphate granules is released earlier than mRNA trapped in fibrin networks [32]. Regarding mesh size, smaller pores and denser structures slow down mRNA release [31,34]. In addition, pore size may condition the chronic inflammation generated by the scaffold [28], a process that directly influences TAM performance. 

Matrices can be produced using multiple methods such as electrospinning, 3D printing, fibre mesh, injection moulding, gas foaming, freeze-drying, and others [40]. Especially, freeze-drying has been commonly used to produce scaffolds for tissue engineering applications [40]. Although some parameters can be adjusted during these processes, these techniques still need further improvement to allow precise control of pore size and geometry, as well as pore distribution.

Because of their biocompatibility, biodegradability, and cell adhesion capability [41,42], natural extracellular matrix (ECM)-derived materials such as collagen and fibrin have been widely studied as mRNA-delivery platforms [20,21,38]. However, they have drawbacks due to their low mechanical strength and rapid biodegradation that limit their use [38].

Collagen is the major constituent of the ECM and one of the most used materials in the design of TAMs [9,20,23,31,33,37,43,44,45,46,47]. This material can be processed in diverse forms, such as sponges, membranes, and fibre matrices [31,43,47,48]. This form has a direct effect on therapy effectiveness. For example, aligned nanofibrillar scaffolds direct tissue formation [33,49] and favour the development of spatially patterned structures such as blood vessels [50]. Moreover, these types of scaffolds can modulate tissue morphogenesis and cellular reprogramming [51,52].

To obtain collagen TAMs with improved biomimetic properties for bone regeneration, the addition of nanohydroxyapatite has been studied. Although the influence of nanohydroxyapatite on transfection efficiency was not tested, Wang et al. [37] reported that this material increases the mechanical strength of the scaffolds without affecting cell growth. Conversely, an excessive amount of nanohydroxyapatite reduces its porosity, which could impair vascularization and fluid permeability, ultimately hindering host cell growth, migration, and proliferation [53]. 

Likewise, the chemical cross-linking of collagen–glycosaminoglycan scaffolds can also reinforce the matrix and decrease its enzymatic degradation [37,54]. Moreover, it favours an increase in the number of osteoblasts within the network and a better distribution of these cells [54]. These considerations contrast with a study by Hu et al. where low-stiffness scaffolds showed improved osteogenic differentiation in vitro and better results in vivo [55]. Studies performed with pDNA matrices have resulted in some contradictory information regarding the effect of stiffness. While it has been stated that low-stiffness scaffolds increase pDNA transfection [56], Ledo et al. performed a systematic study where they observed a significant increase in transfection with the stiffer prototypes [57]. However, whether this tendency of pDNA is translated to mRNA is still unknown. In addition, it has been reported that the mechanical properties of the matrix can direct cell differentiation [58]. Therefore, they should be tailored to the target tissue. Although chemical cross-linking is commonly utilized, physical methods such as those based on electrostatic interactions can also be exploited [59]. 

Fibrin and fibrinogen hydrogels have also been explored as TAMs [21,22,32,34] since both are used in tissue engineering applications due to their role as the primary healing scaffold of the body [60]. These materials create 3D fibrous networks that allow, on the one hand, the release of entrapped substances and, on the other hand, cell colonization. However, there are differences in the behaviour of fibrin and fibrinogen matrices. Low thrombin content hydrogels (i.e., fibrinogen) generate fewer cross-linked networks, with faster diffusion of entrapped molecules, including mRNA complexes [34,61]. This fast diffusion of fibrinogen scaffolds results in fast mRNA release [34]. In addition, Fayed et al. determined that fibrinogen was better suited than fibrin for the fabrication of transcript-activated coatings on titanium implants [34]. This was because fibrinogen coatings exhibited better transfection efficiency, which could be related to improved cell attraction and adhesion [62]. Indeed, it was observed that fibrinogen has strong adsorption onto titanium surfaces [63].

Other studies have supported the importance of fibrin content in TAM performance, as it conditions the rate of matrix degradation [22]. Ledo et al. observed that 2 mg/ml fibrin TAMs degraded earlier than those of 4 mg/ml [22]. When those TAMs, activated with a chondrogenic SOX9 mRNA, were tested in mesenchymal stem cell cultures (MSCs), no differences in transgene expression levels were observed. However, the 2 mg/ml TAMs showed higher expression of other chondrogenic markers than the 4 mg/ml matrices and higher ECM synthesis. These results were attributed to a faster matrix degradation with consequent faster cellular remodelling. The same trend was seen for immature myogenic markers in TAMs activated with MYOD mRNA, although in this case, mature myogenic markers were more expressed in 4 mg/ml fibrin matrices. 

Finally, we can mention Matrigel™ as another protein-based ECM-derived material utilized for TAM preparation [24]. Still, its use is not as widespread as those compounds mentioned before.

As for natural polymers derived from polysaccharides, we can highlight chitosan–alginate hydrogels. One of the main advantages offered by this combination is that they form injectable hydrogels that reduce TAM therapy invasiveness [36]. This, added to the capacity of these materials to activate dendritic cells (DCs) and promote DC co-stimulatory expression markers [64], makes them of special interest for the design of TAMs with immunomodulatory properties [27]. Likewise, the combination of these compounds allows modulation of the mRNA release rate. This is because the chitosan provides positive charges that favour mRNA retention in the matrix. Conversely, matrices composed of alginate show a faster release of mRNA due to the repulsive forces between the negative charge of the matrix and that of mRNA [36]. 

Of note, highly biocompatible polynucleotide-only TAMs have also been prepared. Concretely, these TAMs are based on pH-responsive nano-hydrogels whose only components are DNA and coding mRNA [48]. This composition remains highly experimental at this point, and further data are needed to gauge its medical potential. 

Despite the advantages offered by these natural polymers, their purification is complex, and traces of RNAses and other enzymes can degrade the therapeutic mRNA [49]. Furthermore, there is a risk of immunogenicity and disease transmission [65]. Consequently, the use of synthetic polymers such as poly(lactic acid) (PLA), poly(lactic-co-glycolic acid) (PLGA), or hydroxyethyl methacrylate (HEMA) has also been tested [28,34,66]. One of the main benefits of synthetic materials is that they allow good physical–chemical control of their properties [67]. However, they lack the intrinsic cell-attractive capacity of natural polymers, and thus, they benefit from further modifications in their structure [65].

Both PLA and PLGA are synthetic polymers with a large pharmaceutical record and are widely used in regenerative medicine [68]. Contrary to most natural polymers, they possess high mechanical stability [69]. When these materials form mRNA-activated coatings, their concentration can influence the rate of mRNA release and the effectiveness of the therapy. It has been reported that lower PLA concentrations improve transfection efficiency [34]. This could be because thinner coatings are produced, which allows faster mRNA release and, consequently, better cellular uptake and transfection. However, excessively rapid release can lead to increased toxicity [34]. 

These synthetic materials can also enhance the characteristics of products already used in clinical practice. For example, PLGA microparticles added to a calcium phosphate injectable bone defect filler can improve cell ingrowth by increasing the porosity of the cement [66]. In addition, the microparticles allowed the loading of mRNA nanoplexes into the cement without losing their bioactivity, acting as intermediary matrices for mRNA release [66]. As a result, two bioactive elements, mRNA and calcium phosphate cement, are united in this composite system.

In a similar approach, Balmayor et al. used calcium phosphate granules activated with BMP-2 mRNA to promote osteogenesis in bone marrow-derived stem cells (BMSCs) [32]. However, in this case, lipoplex activity was maintained after incorporation into the granules without the need for an intermediate matrix. They also showed that the combination of mRNA with this bioactive matrix led to a synergistic effect with increased expression of osteogenic markers collagen type-I and osteocalcin and increased mineralization in vitro in comparison to mRNA-activated fibrin gels.

Mineral microparticles have also been used as TAMs due to their ability to sequester proteins. Khalil et al. observed that the efficacy of basic fibroblast growth factor (bFGF) mRNA therapy could be increased when it is delivered from mineral-coated hydroxyapatite microparticles [70]. Once the mRNA is released and expressed, the growth factors produced are retained in the matrix and released in a sustained manner, prolonging their therapeutic effect. This would make it feasible to reduce the dose required and, consequently, the side effects. This ability to retain expressed proteins has also been verified in collagen scaffolds [44].

Overall, these studies show the wide variety of materials that have been investigated as support matrices in TAM design and how different features such as cell binding, biodegradability, or porosity can modulate the final response of the system.

### 2.3. mRNA Modifications

Among the main shortcomings that have slowed down the adoption of mRNA therapeutics, the most important have been its instability, immunogenicity, and low translation efficiency [43]. By chemically modifying synthetic mRNA, transcripts with enhanced bioactivity and lower immunogenicity can be obtained. The most common changes are made in the 5′ cap, the 3′ poly(A) tail, and the nucleotides (Figure 3).

It is known that the incorporation of an anti-reverse cap analogue (ARCA) into the 5′ cap can significantly increase translation in comparison with the use of conventional cap analogues [71]. This is because, in ARCA, the 3′ OH group (closer to 7-methylguanosine) is replaced with -OCH3, so RNA polymerase can only initiate transcription with the remaining OH group. In this way, ARCA incorporation is ensured only in the forward orientation and 100% of the synthesized transcripts can be translated [71]. 

Protein production can also be increased using the incorporation of a translation initiator of a short 5′ UTR (TISU) sequence [34,43]. The 5’ UTR region is a non-coding region of the mRNA that modulates translation initiation. This region is rich in complex secondary and tertiary structures that can be modified by protein binding. Consequently, ribosomal recruitment may be regulated [72]. The 5’ UTR region also has linear structures such as the Kozak sequence. It has been reported that a strong Kozak sequence enhances start codon recognition and increases mRNA translation [72]. Other non-coding regions of the mRNA (i.e., the 3’ UTR region) can also modulate translation.

Another widely employed modification occurs in the poly(A) tail. When its length is extended to 120 bp or more, the stability and translation of the mRNA are improved [73]. 

Surprisingly, the restriction enzymes used to linearize the DNA plasmid during chemically modified mRNA (cmRNA) in vitro production may also affect translational efficiency. For instance, the use of NotI in the pVAXA120 vector to produce BMP-2 cmRNA has led to a 4-fold increase in BMP-2 production when compared to XbaI [21]. 

As for immunogenicity, nucleotide substitutions in the mRNA sequence can significantly reduce it, also providing better stability [20,74]. In particular, the incorporation of modified pyrimidines replacing uridine and cytosine prevents the interaction of RNA with toll-like receptors and retinoid-inducible gene 1 in human peripheral blood mononuclear cells, thus reducing inflammation [74,75]. Some common modified bases include pseudouridine, 2-thiouridine, N1-methylpseudouridine, 5-iodo-uridine, and 5-methylcytidine or 5-iodo-cytidine [33,34,45]. It has been reported that the simultaneous substitution of cytidine and uridine for 5-methylcytidine and pseudouridine improve cell viability and protein expression over substitutions of only one of these nucleotides [76]. Moreover, these double substitutions can be made on all nucleotides or only part of them. Although most TAMs use cmRNA with partial substitutions of both pyrimidines [31,32,34,43], Elangovan et al. observed that complete substitution of both pyrimidines with 5-methylcytidine and pseudouridine did not trigger the innate inflammatory response in BALB/c mice, in contrast to partial substitutions of 25% or unmodified mRNA [20].

Taken together, these results show the advantages the cmRNA possesses compared with wild-type mRNA. Of note, the performance observed with cmRNAs in vitro may differ from the results obtained in vivo. For example, Zhang et al. created a BMP-2 transcript that elicited lower cytokine production by human peripheral blood mononuclear cells (hPBMCs) in vitro compared to unmodified mRNA [43]. However, when this mRNA construct was applied in vivo, it produced an elevated level of circulating cytokines, especially IL-1α [9].

Chemical modifications can reduce protein–mRNA interactions and affect the effective translation of the mRNA, as well as some advanced sequence functionalities [70]. In addition, they are expensive, and each modification needs to be tailored to the gene and cell type used [37,70]. Therefore, several groups have studied the co-delivery of viral anti-inflammatory proteins with wild-type mRNA as an alternative to the use of cmRNA. This combination was expected to decrease the inflammatory response to the mRNA and, consequently, increase its expression. For example, Khalil et al. created matrices that delivered B18R, a protein that inhibits interferon-γ production, together with wild-type mRNA. With this strategy, they obtained transfection efficiencies similar to those of cmRNA [70]. Along the same lines, Wang et al. [37] incorporated mRNA coding for NS1 into collagen–nanohydroxyapatite matrices. NS1 is an immune evasion protein derived from the influenza A virus that inhibits immune sensing and inflammatory cytokine production. As expected, the co-delivery of NS1 mRNA with BMP-2 mRNA showed significantly higher (2-fold) bone formation in vivo than the use of only BMP-2 mRNA.

### 2.4. Nanosystems Employed in mRNA-Activated Matrices

The delivery of exogenous mRNA into the organism is a complex task. Firstly, mRNA is a negatively charged macromolecule. Thus, its transport across the negatively charged plasma membrane is limited. Once this barrier is overcome, the endosomal system retains and degrades virtually all mRNA, suppressing its bioactivity. In addition, mRNA has a short half-life [77]. Therefore, safe and effective vehicles are needed to improve the delivery of mRNA.

Nowadays, non-viral vectors are widely investigated due to their safety profile compared to viral vectors [15]. Among them, the most used are cationic polymers and lipids that bind mRNA via electrostatic interactions, forming polyplexes and lipoplexes, respectively. In this way, they protect the mRNA against nucleases, increase its stability and cellular uptake, and facilitate endosomal escape [15,36]. As a result, transfection efficiency is improved. We can see a clear example of this in a study by Yan et al., where they quantified the transfection efficiency of chitosan–alginate gels incorporating mRNA complexes and compared them with gels activated with naked mRNA. Although fewer mRNA complexes were released from the matrices compared with naked mRNA, they led to higher protein production [27]. However, due to their charge, these reagents can interact with the cell membrane and induce cytotoxicity [21,28]. To avoid this, several groups have employed polyethylene glycol as a shielding agent [21,24,31,43,78].

Cationic polymers such as polyethyleneimine (PEI) have been considered one of the most effective vehicles for gene delivery [79]. Therefore, their use in TAMs is widespread [20,23,28,31,46,47]. Nevertheless, these materials have different properties when delivering DNA or mRNA. Firstly, it is known that they may show lower efficiency when carrying mRNA as compared to pDNA [28]. This could occur because polymers with large molecular weights have more affinity for mRNA than for pDNA. For this reason, the release of mRNA from the carrier is hindered, and consequently, so is the transfection [28,80]. Secondly, differences in cytotoxicity have also been observed, as PEI appears to be less cytotoxic when forming complexes with mRNA than with pDNA [23,77]. 

To achieve a good balance between transfection efficiency and cytotoxicity, the amount of reagent added per amount of mRNA needs to be controlled. For instance, Elangovan et al. determined that the optimal N (nitrogen) to P (phosphate) ratio (molar ratio of amine groups of transfection reagent to phosphate groups in the nucleic acid backbone) for PEI was 10 [81]. In addition, both the size and the electrostatic charge of the polyplexes can affect carrier effectiveness. Sizes smaller than 150 nm together with positive zeta potential show better tissue diffusion [82] and higher cellular uptake by clathrin-mediated endocytosis [83].

In recent years, the use of lipid-based nanocarriers in RNA therapies has become more widespread. The main reason for this popularity is that they showed better transfection capacity than polymers [28,35]. Specifically, liposomes stand out from other structures because of their properties, such as high loading capacity [44]. However, when we are choosing a vector to integrate into TAMs, this feature may not be as advantageous as expected. The reason lies in the fact that as they store a greater number of mRNA molecules inside, fewer nanoplexes are needed per matrix, and consequently, it is more difficult to achieve their homogeneous distribution in the system [44]. Another disadvantage of lipid systems is their complexity, as liposome formulation usually requires several components, such as structure-forming lipids, helper lipids, cholesterol, PEGylated lipids, and others [31,44]. 

Many TAMs integrate lipid-based commercial reagents such as Lipofectamine™, DreamFect™ Gold, or 3DfectIN™ [22,24,28,32,33,44,45]. The use of customized lipids has also been reported [9,21,31,34,43,66]. As with polyplexes, the amount of reagent must be optimized in each case. For example, for the best results with 3DfectIN™, a reagent:mRNA ratio of 3:1 is used [22]. Likewise, charge and size could also affect transfection efficiency. Although there is no consensus on the influence of size, it seems that smaller lipoplexes give better results [84]. As for the surface charge, it does not seem to be such a determining factor in the usual range tested [85].

To improve the performance of current transfection reagents, hybrid nanoparticles or lipopolyplexes have also been studied [37,86,87]. Wang et al. studied the activity of these vehicles in TAMs. In their study, mRNA was complexed with histidinylated-linear PEI to form polyplexes. Then, these polyplexes were mixed with Lip100 liposomes (described in [88]) to form lipopolyplexes [37]. The lipopolyplexes demonstrated better efficacy in mRNA delivery in vivo than in vitro, just the opposite of Lipofectamine™ lipoplexes. 

As we have described, transfection reagents have been widely used for TAM development as they improve system performance. However, TAMs without any transfection vehicle have also been prepared [48]. For example, Fu et al. designed pH-responsive DNA nano-hydrogels, which release mRNA under pH stimuli. With this approach, similar or even higher amounts of reporter protein were produced than using Lipofectamine™, but less cytotoxicity was observed [48].

Overall, the choice of the transfection reagent will influence the efficacy and cytotoxicity of mRNA therapy. The charge, size, and N/P ratio of the nanoplexes can modulate the performance of the transfection reagent, being the reason why such parameters need to be optimized in the early design phase. 

## 3. Long-Term Stability of mRNA-Activated Matrices

One of the limitations of mRNA therapeutics is the long-term instability of mRNA. However, there are processes capable of increasing the half-life of TAMs, thus allowing their storage for long periods. Among them, we can highlight vacuum-drying and lyophilization techniques. By removing water from the formulations, the stability and half-life of the products are improved [31,89]. For example, treated collagen matrices can be stored at room temperature for months [31,90]. However, these procedures can lead to a loss of mRNA activity [31,37]. Therefore, current research is focused on the effects of these techniques on TAMs.

For instance, Badieyan et al. [31] designed a collagen TAM treated using vacuum drying. Proprietary lipid lipoplexes remained stable within the matrix for at least 6 months at room temperature. Conversely, when PEI polyplexes were used, mRNA functionality was lost. This instability of PEI polyplexes contrasts with other studies where pDNA/PEI complexes remain stable after drying treatments [91,92]. Furthermore, Badieyan et al. observed that the vacuum drying process was necessary to achieve prolonged protein expression, specifically, for up to 11 days. This could be related to the observation that, after vacuum drying, a closed structure is produced, and the mRNA vehicles are trapped in the matrix. Consequently, lipoplexes are gradually released as the matrix is degraded [31]. Changes in pore structure after vacuum-drying were also observed in cement matrices [93]. However, in this case, pore size and porosity increased after treatment because of material degradation.

Studies on freeze-drying suggest that it may not have detrimental effects on matrix functionality [37,47]. Nevertheless, as with vacuum-drying, different responses of transfection reagents were observed after the procedure. For example, Wang et al. [37] demonstrated that while lipopolyplexes maintain their structure after lyophilization, possibly because of their polyplex core, Lipofectamine™ lipoplexes rupture and collapse. This low resistance of Lipofectamine™ lipoplexes over time was also reported by Oude et al. [44]. In their study, the use of the cell-penetrating cationic peptide PepFect14 significantly improved the retention of mRNA activity at 4 °C compared to the use of Lipofectamine™ lipoplexes. Moreover, it maintained mRNA functionality for up to at least 2 weeks at that temperature. Of note, the use of lyoprotectants such as sucrose [31,43] and cryoprotectants [44] helps to maintain the integrity of the system during the dehydration process.

As we have seen, the choice of transfection reagents not only affects transfection efficiency and cytotoxicity but may also influence the long-term stability of TAMs. Hence, further studies are needed to depict techniques that maintain TAM activity for longer periods. Once this is achieved, the resulting dry products might eliminate the need for cold chain supply, while reducing transport and storage costs in mRNA therapies. 

## 4. Applications

Although most of the research conducted on TAMs seeks their application in regenerative medicine and tissue engineering, their use as vaccines has also been investigated. We highlight the main applications of these systems in Figure 4. The key features of the studies available that use TAMs are summarized in Table 1 and are covered in the following Subsections. 

We note that some of the design considerations previously covered (Section 3) were discussed mostly focused on tissue regeneration applications. However, the intended application will ultimately determine the design space of the matrices. For instance, when seeking to enhance tissue/organ regeneration, we are interested in obtaining non-immunogenic TAMs. Conversely, if we are designing vaccines, immunogenicity will be a desirable characteristic.

### 4.1. Bone Regeneration

TAMs have shown great potential in regenerative medicine as they provide a suitable environment for cell adhesion and growth while functioning as depot systems. For this, we can highlight the use of hydrogels due to their tissue-like water content, their injectability, and their tuneable chemical and physical properties [56].

So far, most studies have focused on bone regeneration through the release of bone morphogenetic proteins (BMPs) mRNAs [9,20,21,23,31,32,34,37,43,44,45,46,47,66]. Once the cells are transfected with these mRNAs, they begin to produce BMPs. These proteins induce stem cell differentiation into osteoblasts and chondroblasts, thus promoting osteogenesis and chondrogenesis, respectively [94]. 

TAMs encoding for BMP-2 are some of the most used since this is a well-known growth factor whose use is approved by the FDA [34]. Recently, De La Vega et al. reported better bone healing in critical-sized femoral osteotomies defects in rats after using BMP-2 TAMs compared to matrices with the recombinant protein counterpart [9]. Moreover, mRNA therapy did not induce callus formation as it did in the protein delivery matrix. Other groups have also tested different formulations and doses of BMP-2 mRNA in vivo, obtaining different results. For example, Elangovan et al. noted bone healing at 4 weeks after implanting a collagen matrix with BMP-2 polyplexes in a calvarial bone defect model in rats [20]. In contrast, Geng et al. observed a similar repair to that observed by Elangovan in a cranial critical-sized defect model in rats using BMP-2/VEGF-A (vascular endothelial growth factor-A) lipoplexes in a collagen matrix. However, the osteogenic effect of the TAM was observed much later in this last study, at 12 weeks [45]. This slower rate of regeneration may be due to the lower mRNA dose (10-fold) used compared to the first study. Low doses of BMP-2 mRNA were also applied in a femur defect model in rats, where they observed bone healing in two weeks. In this case, lipoplexes were integrated into a fibrin gel [21]. Despite these results, it is difficult to compare the efficacy of doses since different models were used in the studies. Most of the in vivo studies were performed in critical-sized calvarial defect rat models (Table 1); these models are frequently used since their results are less biased by natural tissue regeneration than those of non-critical-sized defect models. In addition, other animal models such as ectopic models, present certain limitations in terms of recreating the microenvironment of the target tissue. Unfortunately, TAMs still have not been tested in large-animal models, which would be important to gauge their medical potential.

Interestingly, co-administration of VEGF-A and BMP-2 mRNA could promote angiogenesis and bone healing simultaneously [45]. Therefore, increased bone formation was reported in vivo compared to untreated groups and to groups treated with only one of the mRNAs [45]. In addition, the bone formed presented better quality parameters as higher expression levels of osteogenesis-related genes were observed [45].

The osteoinductive properties of BMP-2 TAMs could also be used to enhance the biological features of metallic implants, thus facilitating implant osteointegration [34]. To this end, Fayed et al. developed titanium implant fibrinogen coatings activated with BMP-2 mRNA, which demonstrated mRNA dose-dependent osteogenic potential in vitro [34].

Apart from BMP-2, BMP-7 and BMP-9 have also been tested [44,46,47]. BMP-9 may have greater osteogenic potential than BMP-2 [46]. However, when comparing the performance of BMP-9 TAMs with BMP-9 pDNA-activated matrices, no significant differences were found in bone formation in vivo [47].

To the same end, osteoinductive mRNAs (Oi-mRNAs) derived from osteogenically pre-differentiated MSCs were embedded in demineralized bone matrix scaffolds. It has been reported that scaffolds with Oi14-mRNAs (obtained at day 14 of the osteogenic differentiation) may result in more collagen and ECM production by MSCs than other Oi-mRNAs [23]. Moreover, in vivo studies suggest that these Oi14-mRNA matrices could promote cell infiltration and bone repair [23].

### 4.2. Other Regenerative Applications

Although there are few studies, TAMs have also been investigated for cartilage and muscle formation [22], wound healing [70], and vascular regeneration [24,33,45].

Pivotal transcription factors such as SOX9 and MYOD are known to direct tissue specification. For this reason, Ledo et al. incorporated mRNAs coding for these factors into fibrin gels [22]. This approach led to tissue specification in vitro, thus promoting chondrogenesis and myogenesis, respectively [22]. Furthermore, TAMs achieved a higher and faster transcription factor expression compared to pDNA-activated matrices. 

We can also mention the use of basic Fibroblast Growth Factor (bFGF) TAMs to treat wounds in murine diabetic models [70]. As observed for BMP-2 [9,45], treatment with bFGF TAMs resulted in better performance than matrices with the counterpart recombinant proteins. 

Finally, to promote vascular tissue formation, endothelial growth factor (VEGF) [24,45] and hepatocyte growth factor [33] TAMs were designed. In the first case, VEGF-A mRNA was embedded into Matrigel™, and then human Isl1+ cardiovascular progenitors were implanted into the gel. It has been observed that VEGF-A has an effect in driving endothelial specification of these cells both in vitro and in vivo [24]. Consequently, this could be an interesting approach to regenerating cardiac vascular tissue, for example, after a heart attack. In the second case, hepatocyte growth factor mRNA delivered from nanofibrillar scaffolds promoted vascular regeneration in a porcine peripheral arterial disease model [33].

### 4.3. Vaccination and Immunomodulation

Apart from regenerative purposes, recent studies have evaluated the immunomodulatory capacity of TAMs [27,28,36]. In the last years, improvements in the design of mRNA therapies led to their clinical application in the SARS-CoV-2 vaccines [95]. One of the main advantages offered by mRNA vaccines is that they allow rapid scale-up in production and elicit stronger immune responses than other forms of gene therapy and some protein treatments. In addition, they can code for multiple antigens simultaneously and are safer than DNA vaccines [96]. Likewise, the mRNA itself acts as an adjuvant through the activation of toll-like receptors (TLR) 7/8 and 3 [28]. Consequently, both humoral and cellular immune responses are activated [97]. However, they still present limitations that hinder their use, such as their rapid degradation via nucleases, their instability during storage, as well as their side effects, and the limited transfection of antigen-presenting cells [28,95,98].

As we have already seen, the formation of mRNA complexes with non-viral vectors improves the stability and cellular uptake of mRNA [15,36]. Nevertheless, these materials could attenuate the intrinsic immunogenicity of the mRNA, thus hindering its activity as a co-adjuvant [27]. To improve delivery, mRNA nanocarriers have been incorporated into matrices. On the one hand, the matrices allow a localized delivery and a biphasic release of mRNA nanoplexes [36], which could eliminate the need for applying booster doses [27]. On the other hand, subcutaneously implanted TAMs improve cellular uptake and in situ protein expression in comparison with a bolus injection [28]. The matrices could also enhance the transfection of dendritic cells (DCs) by mediating their recruitment and activation within the scaffold [28,29]. For example, Chen et al. [28] demonstrated that chitosan-alginate gels can attract DCs that, once activated, migrate to lymph nodes to initiate the adaptive response. Consequently, scaffold-mediated mRNA delivery produced a robust T-cell response at 5 days post-immunization in vivo. Unfortunately, the humoral response was lower than with protein-based immunization, despite being faster. Similarly, Dastmalchi et al. incorporated CXCL9 cytokines into TAMs to promote immune cell trafficking. After matrix injection into murine tumour models, an increase in the recruitment of DCs as well as the upregulation of NK cells were observed. By delivering both tumour mRNAs and CXCL9, they observed a significant improvement in the survival of the animals compared to the control group [29].

Overall, these studies support the potential for TAMs as immunomodulatory platforms capable of improving the performance of mRNA vaccines. Moreover, due to their characteristics, they would be of special interest in cancer treatment since they could combat tumour immunosuppression [27,28,29]. However, formulations still need to be optimized to strengthen the response generated. This could be achieved using co-delivery of mRNA with proteins, adjuvants such as TLR agonists, or cytokines to attract DCs and other immune cells [27,29,99]. However, this poses technological challenges as the delivery platforms need to be adapted to different payloads. 

## 5. Conclusions and Future Perspectives

Although there are still limited studies on mRNA-activated matrices, those that do exist show promising results. Their use in tissue engineering and immunomodulation may offer alternatives to recombinant protein and pDNA therapies, and many indicate more stronger performance than those technologies. 

Key design parameters of mRNA-activated matrices directly affect their therapeutic activity. Firstly, chemical modifications in the mRNA sequence have improved mRNA stability and translation while reducing its immunogenicity. Secondly, the composition and structure of the matrix strongly influence cellular adhesion and proliferation as well as the kinetics of mRNA release.

As for the cytotoxicity of TAMs, although biocompatible biomaterials have been used, the incorporation of transfection reagents can generate or exacerbate side effects. Therefore, it is still necessary to keep investigating to identify vehicles that combine high transfection efficiency and low toxicity. Recently, the use of biological vehicles such as exosomes, extracellular vesicles that carry endogenous mRNA, has started to gain attention in regenerative medicine and those have even been added to polymer matrices. However, there is no control over the mRNA sequence that is delivered, thus representing a different strategy from TAMs. 

Moreover, the choice of vectors will also influence the long-term stability of the TAMs. The need to produce off-the-shelf products that can be shipped globally has led to further research into techniques that improve TAM stability. This, however, is also a matter that remains to be solved.

## Figures and Tables

**Figure 1 jfb-14-00048-f001:**
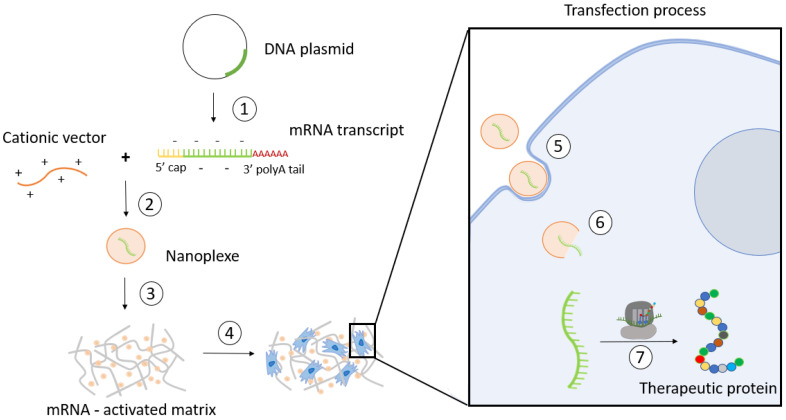
Schematic illustration of TAM preparation and the mechanism of action: (1) in vitro transcription; (2) formation of nanoplexes; (3) incorporation of nanoplexes into the matrix; (4) cell colonization of the matrix; (5) internalization of nanoplexes; (6) mRNA cytoplasmic release; and (7) mRNA translation.

**Figure 2 jfb-14-00048-f002:**
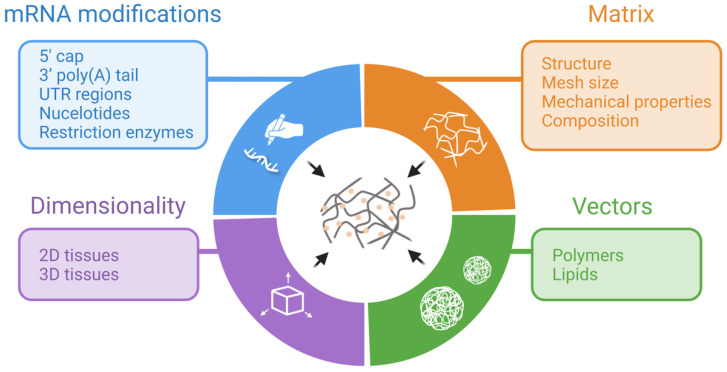
TAM design considerations. Created with BioRender.com on 27 December 2022.

**Figure 3 jfb-14-00048-f003:**
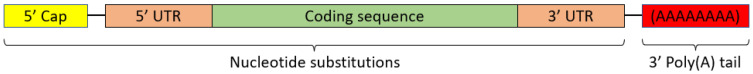
Structure of mRNA indicating regions critical for translation and possibilities for optimization and chemical modification.

**Figure 4 jfb-14-00048-f004:**
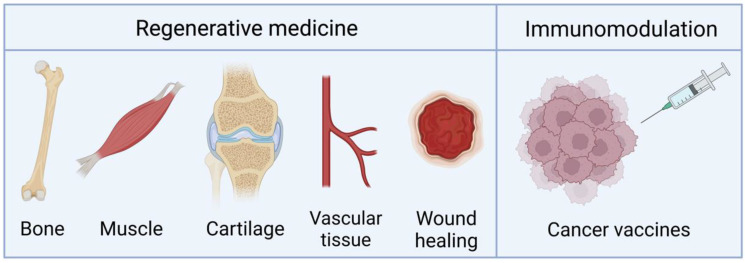
Main applications of TAMs. Created with BioRender.com on 27 December 2022.

**Table 1 jfb-14-00048-t001:** Published studies about TAMs.

Application	Matrix Composition	Encoded Protein	mRNA Modifications	Transfection Reagents	Development	Publication Year
Bone regeneration	Collagen scaffold	BMP-2	ARCA	PEI	In vitro and in vivo (rat critical-sized calvarial bone defect model)	2015 [20]
			s2U (0.25) m5C (0.25) or Ψ (1.0) m5C (1.0)			
			Poly(A) - 120			
Bone regeneration	Collagen scaffold	BMP-7	Cap 1 structure	PepFect14 or Lipofectamine™ MessengerMAX	In vitro	2022 [44]
			Sequence modifications			
Bone regeneration	Collagen scaffold	BMP-2 or BMP-9	ARCA	PEI	In vitro and in vivo (rat critical-sized calvarial defect model)	2017 [46]
			Ψ (1.0) m5C (1.0)			
			Poly(A) - 120			
Bone regeneration	Collagen fibre matrix	BMP-2 and VEGF-A	m1Ψ (1.0)	Lipofectamine™ MessengerMAX	In vitro and in vivo (rat critical-sized calvarial defect model)	2021 [45]
			Poly(A) tail			
Bone regeneration	Collagen-nanohydroxyapatite matrix	BMP-2 and NS1	Poly(A) tail	Lipopolyplexes His-lPEI/Lip100 or Lipofectamine™ MessengerMax	In vitro and in vivo (mouse ectopic model)	2021 [37]
Bone regeneration	Collagen sponge scaffold	BMP-2	Cap structure	Proprietary lipid/DPPC/cholesterol/DMG-PEG or PEI	In vitro and in vivo (rat non-critical femoral bone defect model)	2016 [31]
			s2U m5C			
			Poly(A) - 200			
Bone healing	Collagen sponge scaffold	BMP-2	TISU sequence	Proprietary lipid/DPPC/cholesterol/DMG-PEG	In vivo (rat critical-sized femoral osteotomies defect)	2022 [9]
			5IU (0.35) 5IC (0.075)			
Bone healing	Collagen sponge scaffold	BMP-2	ARCA	Proprietary lipid/DPPC/cholesterol/DMG-PEG	In vitro and in vivo (rat critical-sized femoral defect)	2019 [43]
			TISU sequence			
			5IU (0.35) 5IC (0.075) or s2U (0.25) m5C (0.25)			
			Poly(A) - 120			
			Poly(A) tail			
Bone regeneration	Perforated collagen membranes	BMP-9	ARCA	PEI	In vitro and in vivo (rat critical-sized calvarial defect model)	2019 [47]
			Ψ (1.0) m5C (1.0)			
			Poly(A) - 120			
Bone regeneration	Demineralized bone matrix scaffold	Oi-mRNA	None	PEI	In vitro and in vivo (rat critical-sized calvarial defect model)	2021 [23]
Bone healing	Fibrin gel	BMP-2	ARCA	Proprietary lipid/DOPE/cholesterol/DMPE-PEG	In vitro and in vivo (non-critical rat femur bone defect model)	2016 [21]
			s2U (0.25) m5C (0.25)			
Bone regeneration	Fibrin gel or micro-macro biphasic calcium phosphate (MBCP) ceramic granules	BMP-2	ARCA	DreamFect™ Gold	In vitro	2017 [32]
			s2U (0.25) m5C (0.25)			
			Poly(A) tail			
Bone healing	PLGA microspheres in calcium phosphate cements	Reporter proteins	s2U m5C	Proprietary lipid/DOPE/cholesterol/DMG-PEG	In vitro	2017 [66]
			Poly(A) - 200			
Ortho-regeneration	Poly-D,L-lactic acid (PDLLA), fibrin or fibrinogen coating	BMP-2	ARCA	Proprietary lipid/DPPC/cholesterol/DMG-PEG	In vitro	2021 [34]
			5IU (0.35) 5IC (0.075)			
			Poly(A) - 200			
Tissue engineering	Chitosan-alginate hybrid hydrogels	Reporter proteins	ARCA	GenaxxoFect™ reagent	In vitro	2018 [36]
			Ψ (1.0) m5C (1.0)			
mRNA delivery	DNA nano-hydrogel	Reporter proteins	m7G cap	None	In vitro	2021 [48]
			Poly (A) tail			
Chondrogenesis and myogenesis	Fibrin gel	SOX-9 or MYOD	ARCA	3DfectIN™	In vitro	2020 [22]
			Kozak consensus sequence			
			alpha-globin 3′ UTR terminating			
Vascular regeneration	Matrigel™	VEGF-A	ARCA	Lipofectamine™ RNAiMAX	In vitro and in vivo (NOD/SCID mice)	2013 [24]
			Ψ m5C			
			Poly(A) tail			
Vascular regeneration	Parallel-aligned nanofibrillar collagen scaffolds	HGF	Cap 1 structure	Lipofectamine™ Messenger Max	In vitro and in vivo (porcine peripheral arterial disease model)	2020 [33]
			Ψ m5C			
			Poly(A) - 175			
Wound healing	Mineral-coated microparticles (MCMs)	bFGF	ARCA	Lipofectamine™ Messenger Max	In vitro and in vivo (murine model of diabetic ulcers)	2020 [70]
			Ψ m5C			
			Poly(A) tail			
Vaccine	pHEMA scaffold	Reporter proteins	m7G cap	Lipofectamine™ Messenger Max or Stemfect™ or in vivo-jetPEI™ or Poly (β-amino ester))	In vitro and in vivo (mouse subcutaneous implant model)	2018 [28]
			Poly(A) tail			
Vaccine	Chitosan-alginate 3D porous gel	OVA	m7G cap	Stemfect™	In vitro and in vivo (murine model)	2018 [27]
			poly(A) tail			
Vaccine	Hydrogel	Tumour proteins	None	Nanoparticles	In vitro and in vivo (murine glioblastoma multiforme model)	2021 [29]

Abbreviations: Ψ: pseudouridine; 5IC: 5-iodo-cytidine; 5IU: 5-iodo-uridine; ARCA: 3′-O-Me-m7G(5′)ppp(5′)G RNA cap structure analog; bFGF: basic fibroblast growth factor; BMP: bone morphogenic protein; DMG-PEG: 1,2-dimyristoyl-rac-glycero-3-methylpolyoxyethylene PEGylated lipid; DOPE: dioleoylphosphatidylethanolamine; DPPC: dipalmitoyl phosphatidylcholine; HGF: hepatocyte growth factor; m1Ψ: N1-methylpseudouridine; m5C: 5-methylcytidine; m7G: 7-methylguanosine; OVA: ovalbumin; PEI: poly-ethylenimine; pHEMA: hydroxyethyl methacrylate; s2U: 2-thiouridine; VEGF-A: vascular endothelial growth factor–A.

## Data Availability

No new data were created or analyzed in this study. Data sharing is not applicable to this article.

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
