# Peer review of "Matrices Activated with Messenger RNA"

_jfb, 2023, doi:10.3390/jfb14010048_

Round 1
Reviewer 1 Report
This review is well-written and organized. The authors presented the background, current knowledge, and potential applications for mRNA-activated matrices. All presented figures are attractive and help audiences understand the content better. This review is very interesting and can be accepted for publishing in this journal after minor revision.
Comments:
- limitations/disadvantages/challenges should be summarized and presented.
- Table 1 is very interesting and informative but it is really difficult to read (too many columns). The author may consider to simplify this table e.g. deleting the reference column and put the reference in year column. Or this table could be presented in landscape orientation.
- In Long-term stability of mRNA–activated matrices sections, Besides Badieyan et al. [30], the authors should provide more example regarding the topic to provide more evidences.
Author Response
First of all, we would like to thank you for your contribution to the review. We have incorporated those suggestions in our review, and more concretely:
- We have included information regarding the limitations of TAMs (see pag. 3, "However, these delivery platforms still have some limitations...").
- We have simplified Table 1 by combining year and reference as suggested by the reviewer.
- We have added further information on the effect of vacuum-drying treatment on polyplexes/matrices. Concretely this in pag. 11 ("For instance, Badieyan et al. [31] designed...").
Reviewer 2 Report
Dear Author, please answer the following comments in your manuscript
For a review article, 3 figures seem very less. Please include additional figures/ schematic diagrams in appropriate places.
Include a brief paragraph on how mRNA are used for cellular reprogramming when stem cells are used for tissue engineering applications.
Give a brief note on how exosomes/extracellular membrane vesicles are used as a carrier for mRNA delivery and their applications in tissue engineering.
Author Response
First of all, we would like to thank the reviewer for the constructive comments, which we have tried to implement in the manuscript to our best understanding. We summarize below how we have tied to address those:
- We have included a new figure in the manuscript (figure 3) to illustrate mRNA regions and possibilities for modification.
- We have included information on the use of mRNA in cellular reprogramming. This is in pag. 3 "These properties have led TAMs to be investigated ...".
- Although the main focus of the review is on matrices loaded with carriers having mRNA with well defined sequences, we have made a mentioned to the use of exosomes (in pag. 17 "Recently, the use of biological vehicles...").